# Experimental Security Analyses of Access of Browser Extensions Accessing Sensitive Input Fields

## ABSTRACT

Browser extensions offer a variety of valuable features and functionalities. They also pose a significant security risk if not properly designed or reviewed. Prior works have shown that browser extensions can access and manipulate data fields, including sensitive data such as passwords, credit card numbers, and Social Security numbers. In this paper, we present an empirical study of the security risks posed by browser extensions. Specifically, we first build a proof-of-concept extension that can steal sensitive user information. We find that the extension passes the chrome webstore review process. We then perform a measurement study on the top 10K website login pages to check if the extension access to password fields via JS. We find that none of the password fields are actively protected, and can be accessed using JS. Moreover, we found that 1K websites store passwords in plaintext in their page source, including popular websites like `Google.com` and `Cloudflare.com`. We also analyzed over 160K Chrome Web Store extensions for malicious behavior, finding that 28K have permission to access sensitive fields and 190 store password fields in variables. To analyze the behavioral workflow of the potentially malicious extensions, we propose an LLM-driven framework, *Extension Reviewer*. Finally, we discuss two countermeasures to address these risks: a bolt-on JavaScript package for immediate adoption by website developers allowing them to protect sensitive input fields, and a browser-level solution that alerts users when an extension accesses sensitive input fields. Our research highlights the urgent need for improved security measures to protect sensitive user information online.

### ACM Reference Format:
Anonymous Author(s). 2023. Experimental Security Analyses of Access of Browser Extensions Accessing Sensitive Input Fields. In *Proceedings of ACM Conference (Conference'17)*. ACM, New York, NY, USA, 12 pages. https://doi.org/10.1145/nnnnnnn.nnnnnnn

## 1 INTRODUCTION

Browser extensions, while enhancing web browsers and user experience, pose significant security risks. The underlying cause of the risk is the unfettered access of the HTML DOM tree to browser extensions (or any JavaScript) loaded onto the webpage. Extensions are loaded at the same context level as that of the DOM nodes. This leads to a lack of security boundary between the extension and the content of the webpage, including sensitive information that

users may enter. This violates the users' expectation of security with respect to sensitive information such as passwords, credit card information, and Social Security Numbers (SSNs). Guha et al [10] first identified this issue in 2011 and proposed restricting access to potentially sensitive DOM nodes. Subsequently, Liu et al [15] proposed a new permission model by adding a sensitivity attribute to HTML elements to manage access to sensitive elements. These proposals did not become mainstream, possibly due to their impact on the usability of extensions such as password managers. Password managers rely on accessing the password fields to save users' passwords and provide autofill features that do not require users to remember the passwords. Implementing these security measures might impact the usability of the extensions like password managers. As such, these vulnerabilities are still present in the browser ecosystem.

Prior work has shown that it is possible to exploit these vulnerabilities to read sensitive user data such as emails [15, 25], passwords [2, 25], and even perform phishing attacks [2, 20, 25]. These attacks can either use: a) *Static Code Injection* where the attackers add the malicious code in the extension; or b) *Dynamic Code Injection* where the code is loaded dynamically from a remote server and executed at run time. Static code injection is impractical as they can be detected by static code analysis [5, 6, 29, 34]. Dynamic code injection bypasses the static security checks as the code is injected at run time, and thus, is harder to detect [11, 24]. To address this vulnerability posed by dynamic code injection, Google introduced new regulations in Manifest V3 that disallowed the execution of remotely injected code. However, as we show in Section 3.3, it is possible to bypass the defense and execute malicious remote code to steal sensitive information.

In this work, we conduct an empirical study to understand the extent to which these vulnerabilities can be exploited. First, we develop a proof-of-concept extension that extracts users' passwords while being disguised as a ChatGPT plugin (Section 3). We submit the extension to Chrome WebStore and find that it passes the review process, indicating the feasibility of such an attack. Next, we analyze the login pages of the top 10K domains to see if the password values can be extracted using our extension. Finally, we perform static and dynamic analysis of 19K extensions on the WebStore to identify the following: (1) extensions that have the necessary permissions to carry out the attack; and (2) extensions that are actively accessing and storing password values. With this analysis, we identify 190 potentially malicious extensions that access password fields.

To further understand the behavioral data flows of these malicious extensions, we propose, *Extension Reviewer*, a novel LLM-driven framework that helps review the extensions. We use LLMs because, while static and dynamic analysis effectively analyze code structures and patterns, they do not capture behavioral patterns that emerge from high-level logic. LLMs have been shown to have richer and more detailed high-level understanding, allowing them

to complete tasks like generating unit tests for software [22, 27] and reasoning about program invariants [19]. Furthermore, static and dynamic analysis can be sensitive to code obfuscation [1].

In the *Extension Reviewer* framework, LLMs are tasked to analyze the source code of the extensions and use chain-of-thought prompting to understand the behavioral flows of the extension. Performing this analysis, we identify one extension that is accessing the password fields and sending the passwords over the network. We further show that *Extension Reviewer* can identify potentially malicious extensions even when the malicious execution is dynamically loaded at runtime. We plan to release the framework publicly.

Finally, we discuss two countermeasures to mitigate the security risks from the observed vulnerabilities. In our *Bolt-on* solution, we provide a JavaScript package that website developers can adopt today to mitigate the attacks (Section 8.2). The package introduces a new input type *SecureInput* that uses WeakMaps[1] to store sensitive values in private variables. We also discuss a more fundamental browser-level solution (Section 8.2) by instrumenting chromium to alert users when an extension accesses sensitive input fields. We also discuss the impact of these solutions on password managers and argue that the usability of password managers can be maintained without compromising on the security of the input fields.

**Contributions.** In this work, we make the following contributions:

- We develop a proof-of-concept browser extension disguised as a ChatGPT plugin, demonstrating that it can bypass the Chrome WebStore review process, thereby highlighting potential weaknesses in the current review mechanisms.
- We analyze the login pages of the top 10K domains, revealing that many websites are susceptible to potential attacks from malicious extensions. Our analysis of 19K extensions on the Chrome WebStore further identified that a significant number have the necessary permissions to exploit these vulnerabilities.
- We introduce *Extension Reviewer*, a novel LLM-driven framework designed for in-depth browser extension source code analysis. This tool, enhanced by chain-of-thought prompting, can effectively identify extensions that access sensitive user data and detect dynamically loaded malicious code.

## 2 BACKGROUND AND RELATED WORKS

### 2.1 HTML Fundamentals

**HTML Input Elements:** Input fields, marked by the <input> tag, serve as the most basic avenue for users to input data into a webpage. Password fields, generally used for sensitive content, obfuscates the text written in the input field. We note that ensuring that input fields cannot be accessed by malicious actors is crucial, as exposed sensitive data can be harvested by automated scripts or bots.

**DOM Tree:** While rendering a webpage, the browser constructs a Document Object Model (DOM) of the page. This DOM, composed of nodes and objects, replicates the webpage as a tree structure, known as the DOM Tree. The tree's root initiates with the <html>

element. The nodes of the tree can be accessed, and manipulated by any JavaScript (JS) loaded on the page via the DOM API.

**Dynamic Code Injection:** JavaScript allows the execution of strings as JavaScript code using the eval() function. While eval can be legitimately used to generate code based on specific conditions dynamically, its use is generally viewed as a security risk due to its potent nature. Extensions have been known to use eval statements to inject code into webpages dynamically. Kapravelos et al. [11] found more than 400 Chrome extensions using eval statements with inputs exceeding 128 characters in length. Similarly, Wang et al. [28] discovered 145 extensions on the Firefox add-on store that contain the eval statement. Subsequently, Google introduced *Manifest V3* removing the usage of *eval* statements.

**Browser Architecture:** Prior research has investigated how modifications to browsers' underlying structure can enhance user privacy and security. Louw et al. [16] suggested incorporating a new runtime monitoring framework to observe an extension's access to sensitive APIs, such as adding an event listener to secure fields like passwords. Guha et al. [10], and Liu et al. [15] recommended adding new permissions to access specific DOM elements. Bauer et al. [2] explored how extensions could bypass the existing Chrome permission structure to execute a range of attacks.

### 2.2 Browser Extensions

**Permission Models:** Browser extensions request permissions for the resources they require for their functionality via the manifest file. Permissions can be of two types: Host permissions and API permissions. Host permissions enable extensions to inform the browser about the websites they need to access, allowing extensions to access content from these specified sites. API permissions, on the other hand, provide extensions with the capability to interact with WebExtension APIs, such as browser.storage or browser.cookies.

**Content Scripts and Background Pages:** Extensions consists of two main components: content scripts and background pages (or service workers). Content scripts are static JavaScript files that are automatically loaded with a webpage. These scripts run in the webpage context as an extension to the DOM tree. Background pages, in contrast, are not loaded with each website; they react to browser events or carry out WebExtension API-based actions. Although content scripts have access to certain WebExtension API functions, their access is limited in scope. To leverage the full extent of the APIs, content scripts communicate with the background page via message passing[2].

While extensions can load static JavaScript as content scripts, they can also use a mix of host permissions and browser APIs to inject JavaScript into webpages programmatically. For example, an extension can request no websites under content scripts but then request scripting and host permissions on all websites to inject content scripts on websites dynamically. Furthermore, content scripts, without host permissions, must comply with website-defined cross-origin restrictions, unlike scripts injected via host permissions and API. This compliance limits their interaction with

---

[1]https://developer.mozilla.org/en-US/docs/Web/JavaScript/Reference/Global_Objects/WeakMap

[2]https://developer.mozilla.org/en-US/docs/Web/API/Window/postMessage

external entities, although they can still send and receive messages from the extension's background script.

**Attacks involving Extension:** Prior studies [7, 16, 17, 28] have detailed various techniques by which malicious extensions could leak sensitive information. Bauer et al. [2] developed an iframe-based attack to stealthily steal user credentials by leveraging the autofill functionality of password managers. Similarly, Perrora et al. [20] crafted an extension that performed an iframe-based phishing attack where their extensions would fetch dynamic codes from a server and execute them. They managed to bypass and publish their extension to the Chrome web store. We note that these attacks are no longer viable due to Chrome's ban on dynamic remote code execution.

In this work, we build a proof-of-concept extension to extract sensitive information, submit the extension to Chrome web store and find that it bypasses the security checks, showing the practicality of the attack (see Section 3.3).

**Detection of Malicious Extensions:** Several previous studies have devised tools and frameworks for the detection of malicious extensions. Research conducted in [29, 34] combined both static and dynamic analyses to identify and flag extensions. While Zhao et al. [34] focused on the detection of information leaks via extensions, Wang et al. [29] emphasized tracking DOM changes to identify malicious extensions. Varshney et al. [25] also introduced a static analysis framework for detecting malicious code within an extension. DeKoven et al. [6] identified malicious extensions by flagging users who behave suspiciously on websites, subsequently scanning all loaded extensions for specific threat indicators. Shahriar et al. [23] utilized a Hidden Markov Model to analyze and detect vulnerable and malicious extensions. Toreini et al. [24] created DOMtegrity to monitor and flag malicious DOM changes like 'document.write' or swapping child nodes. Previous research proposed dynamic analysis frameworks that analyze runtime code bases of extensions and match them to set heuristics to flag them as malicious [5, 11].

Our work complements this line of study, offering a security analysis of vulnerabilities affecting browsers and generic solutions to address these vulnerabilities. We note that in this work, we focus on Google Chrome as it is well-documented, and is the most popular browser.

## 2.3 LLMs and Program Analysis

Large language models (LLMs) are tranformer [26] based models that are trained on massive text datasets, allowing them to generate human-like text and engage in natural language conversations. These models can contain billions of parameters; the release of these models has enabled new applications in areas like conversational agents, text generation, and question answering.

**Program Analsis.** Program analysis refers to analyzing a program's source code to identify errors, and potential security vulnerabilities. LLMs have been used to perform program analysis [14, 19, 33]. For example, LLMs have been proposed to understand the behavior of code constructs [18] and generate test cases [22, 27]. In this work, we leverage LLMs to understand the data flows in extensions that access sensitive data. Specifically, we first identify extensions that access user passwords. We then propose an LLM driven framework to analyze the extension to uncover the data

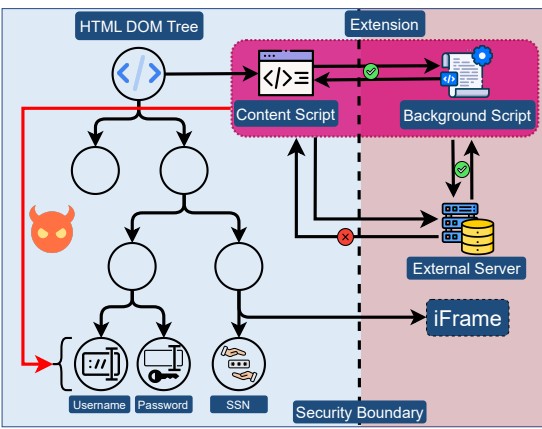

**Figure 1: Comparison between iframes and browser extensions in relation to a website's DOM tree. While iframes are isolated by the Same Origin Policy, extensions operate without such restrictions allowing them to access any element of the DOM tree, including sensitive user data.**

flow associated with the passwords to identify if the extensions are leaking passwords.

**LLM Frameworks.** LLM frameworks are software platforms that enable to users to develop and deploy LLM based applications. These frameworks provide APIs for interacting with LLMs and tools for managing and deploying applications. In this work, we use *LangChain* to build a framework to understand the data-flows in potentially malicious extensions, described in detail in Section 7.

## 3 SECURITY LANDSCAPE OF EXTENSIONS

We analyze the interaction of browser extensions with input fields to identify potential design issues in the accessibility of input fields by extensions. We note here that while JavaScript running on the page can also access the HTML elements in a similar way, we restrict our analysis to extensions in this work as they operate within a controlled environment constrained by browsers' policies which allows us to identify and analyze potential security risks.

### 3.1 Extension Priviledges

The existing permissions framework across all browsers exhibits a coarse-grained approach, particularly with respect to access to web page content. The interaction of extensions with the HTML DOM tree is shown in Figure 1. Once an extension is loaded on a webpage, it has unrestricted access to all elements on the page, including sensitive input fields. Such an extension, essentially a JavaScript program loaded into the DOM tree of the page, can access and potentially manipulate any data in the input fields on the page (Figure 1). This coarse-grained control contrasts with the fine-grained access control for certain software and hardware resources, such as location information or file storage.

One consequence of this coarse-grained model is the absence of a security boundary between the extension and the HTML elements (Figure 1). This contrasts IFRAMES, governed by strict same-origin

policies that restrict access to the parent DOM tree and thus lie outside the security boundary (as shown in Figure 1).

**Isolating Extensions.** Browsers follow a set of rules to isolate extensions to their own environment. Before December 2020, Manifest V2 (MV2) governed the extensions' interactions within the browser's boundaries. Previous research [20] identified MV2's limitations and showed how extensions can bypass it, raising security issues. One significant MV2 security loophole was allowing `eval()` statements, enabling extensions to execute any external JavaScript without any checks. This lead to attacks such as iframe-based phishing and password stealing [20].

## 3.2 Security landscape after Manifest V3

In December 2020, Chrome introduced Manifest V3 (MV3), substantially changing privacy, security, and performance. From a security standpoint, MV3 introduced `declarativeNetRequest` API for network request modifications and discontinued the `webRequest` API, disallowing extensions to modify network requests in real-time, closing a major loophole [9]. MV3 also prohibited the execution of remote code and the use of eval statements. This vulnerability was exploited by attackers in [20] to extract sensitive user data.

Despite MV3's intended advancements in user privacy and security, content scripts' operations remain unchanged. This maintains the lack of security boundary between the extension and web page and allows an extension to be loaded on the DOM tree and gain unrestricted access to the webpage, posing security risks for the users. We use this vulnerability to design our PoC extension (Section 3.3).

**Impact on Review Process.** Before MV3, Chrome's web store review process involved using both static and dynamic analysis, combined with developer-centric heuristics to detect malicious extensions. However, as demonstrated by [20], extensions can bypass this system, enabling the successful upload of a malicious extension to the web store. After MV3, Google prohibited all remote code execution and mandated that all code be included within extensions as this permits more reliable and efficient reviews of extensions submitted to the web store [9].

## 3.3 Building PoC extension

Prior work has exploited the lack of security boundary between the extension and the rest of the DOM tree [3, 7, 17, 25, 28]. They either used static or dynamic code injection to extract sensitive data. Extensions with static code are impractical as the malicious code can be detected via code analysis [6, 10, 25, 29, 34]. On the other hand, dynamic code injection attacks are not feasible after the introduction of MV3. Thus, to build a practical extension and exploit the observed vulnerabilities, we need - a) to access the input elements without using dynamic code injection, b) to overcome any obfuscation on the values of input fields, and c) to submit the extension to chrome web store and clear the review process.

We note here that a malicious extension can also manipulate the elements and modify the content to perform other attacks such as screenshot attack [7] and phishing attacks [2, 20]. However, for the PoC extension, we focus on the security of text input fields and the sensitive information that can be extracted from them. As our primary objective is to build a practical extension to pass the webstore review process and extract sensitive information, we build

a hybrid attack that leverages techniques from static and dynamic code injections. Specifically, we design our extension to include a benign code template that identifies an element with a given *CSS selector*. We dynamically retrieve the *CSS selector* string from a server which allows us to control the input fields at runtime. Once we get access to the sensitive input field, we obtain it's value and store it. This technique is similar to that used by Khandelwal et al. [12]. Note that we do not require additional permission to communicate with the server and retrieve the *CSS selector*. We instead use the background page to fetch the string and pass it through messages to the content script, as shown in Figure 1.

## 3.4 Uploading to Web Store

Finally, we submit the extension to the web store to evaluate the web store's review process. The extension passed the review process on the Google Chrome web store. To hide the extension's malicious aspects, we disguised it as a GPT-based assistant offering ChatGPT-like functions on websites. The extension asked for permission to run on all websites, which is reasonable as most extensions that offer assisting features ask for this permission.

Webstores' failure to identify the malicious extension highlights the need for more robust verification systems for browser extensions. The existing security checks may not be sufficiently comprehensive or effective in identifying potential threats. This is particularly concerning given the potential for extensions to access sensitive user data, including passwords and other input field data, as shown in this work.

**Ethical Considerations.** We maintained ethical integrity throughout the process by adhering to the established guidelines from prior works [20]. Specifically, we ensure we do not collect sensitive information from manual testers during the review process. Our extension was engineered to interact with our servers, identify the type of HTMLElement we were targeting (in this case, input elements), monitor the values on those elements, and ultimately transmit the recorded values back to our server. To protect the privacy of the manual tester while not revealing the extension's malicious nature, we deactivated our data-receiving server, retaining only our element-targeting server online. Consequently, our extension would request the target element, acquire the CSS selector, and then attempt to send the recorded data to a non-existent server. This procedure ensured that the primary operation of the extension remained consistent with our original design. We uploaded the extension to the web store once, ensuring we did not waste testers' time during the manual review process. Additionally, once approved, we immediately removed the extension from the web store. We always kept the extension in "unpublished" mode so the users could not find and install the extension.

Upon approval, we disclosed this vulnerability to Google; however, they responded, stating, *"...We understand that there are malicious Chrome extensions on the store, and it is difficult to limit the number of these extensions."*

## 4 MEASUREMENT OVERVIEW

Our next objective is to conduct comprehensive measurements analyzing the robustness of existing practices in light of the vulnerabilities discussed above. Specifically, we perform large-scale

measurements along two dimensions: 1) Website measurement and Extension Measurement.

**Wesbite Measurement.** In website measurement (Section 5), we analyze the login pages of top 10K websites to check if there are any protections in place for password fields (shown in Figure 2). Interestingly, we find a previously unknown vulnerability, *Plaintext Visible*, where the password values are stored in plain text in the HTML source code of the page. We found that more than 1100 websites had this vulnerability. We also found that password values were accessible via JavaScript APIs for all the login pages that we analyzed. It is noteworthy that the ability of JS to access input fields is essential for various types of form fields and in password managers. However, it also exposes the passwords to any extension that has permission to run on the page.

**Extension Measurement.** In extension measurement (Section 6), we use a combination of static and dynamic analysis to analyze 19K chrome extensions to identify: a) how many extensions have the necessary permission to steal user passwords, and b) how many extensions actively access passwords fields. Figure 4 shows the pipeline for measurement. We find 190 extensions accessing and storing password fields. To further analyze the data flows in these flagged extensions, we propose a novel LLM-driven framework, *Extension Reviewer* (Section 7) that analyzes the source code and allows us to analyze the behavior of the extension. We further discuss the motivation to use LLMs in Section 7.

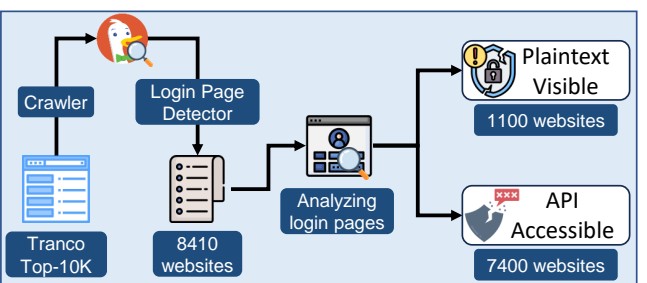

**Figure 2: Our website vulnerability measurement pipeline uses a custom crawler to identify login pages of websites and detect the type of vulnerabilities present.**

## 5 WEBSITES' VULNERABILITIES

We conduct a comprehensive measurement to check the robustness of the password fields against a malicious extension. Our infrastructure consists of a custom-built web crawler to navigate popular websites' login pages and inspect the HTML and JavaScript (JS) elements associated with password fields. The crawler is equipped with capabilities to handle different types of login forms, including both static and dynamic forms. We ran the crawler from a controlled environment to ensure consistency in the measurements.

*Methodology:* We perform the measurement using a Chromium browser controlled via the Selenium library in Python. We also install our PoC extension (Section 3) to extract the passwords. The overview of the measurement pipeline is shown in Figure 2.

We use the top-10K domains from the Tranco list generated on Feb. 2nd, 2023. We employ a two-tiered approach to identify and

analyze the login pages of these domains. First, we attempt to locate the login button on the homepage of each domain by analyzing the text of all clickable elements on the page and searching for keywords associated with the login function. In case of failure, we perform a search on DuckDuckGo using the query `<domain name> log[-?]in`. We then select the top five pages from the search results as potential login page candidates and analyze each candidate page to determine if it is a login page. In particular, we treat a page as a login page if there is a *username/email* field or a *password* field.

After finding the login page, we automatically enter a unique username and password and attempt to extract them using the extension. We note that a login page can exist without password fields (e.g., linkedin.com). Specifically, there can be login pages where the password fields appear only after the email/username is entered. To capture the password field in such cases, we press ENTER after inserting the username and check if the password field is present. This allows us to capture login pages where the password fields are initially hidden.

*Results:* In our study, we identified login pages for 8,410 websites out of the top 10,000 domains. Among these, we found password fields present on 7,140 websites. The remaining 1,270 pages contained username or email fields but no password fields. Notably, we could extract password data from all the websites that presented the password fields. Further analysis revealed that 1,100 websites exhibited *Plaintext Visible* vulnerability; the password values were displayed in plain text within the HTML DOM. Figure 3 shows snapshots of these vulnerabilities, depicting password values in plain text in the HTML. The underlying issue is that the *value* attribute of the *input* element is set to update at each keystroke. In most implementations of password fields, this value attribute is omitted or kept empty.

Notably, we find that the *Plaintext Visible* vulnerability was present on several popular websites, including but not limited to gmail.com and cloudflare.com. The results indicate that this security vulnerability can potentially impact billions of users. The existence of such a basic security oversight on popular websites is concerning, as even websites with substantial resources are not immune to security lapses. We disclose this finding to Google and they responded with "..*We don't consider passwords in HTML to be a serious vulnerability in this case.*"

## 6 EXTENSION MEASUREMENT

**Potential Ability To Exploit Vulnerability.** We analyze the extensions on the Chrome store to identify how many extensions can potentially access sensitive information. We analyze the manifest files and look for extensions that request the `scripting` permission, or that request the content scripts to be run on `all_urls`. Scripting permission allows the extension to inject content script. We find that 12.5% (17.3K) extensions have the necessary permissions to extract sensitive information on all web pages. This includes popular extensions such as *AdBlockPlus* and *Honey* with more than 10M users. We also find that 33.6% (46.4K) extension request content scripts to be run on at least one website.

**Potential Prevalence.** Prior research has demonstrated the existence of malicious extensions in the webstore [7, 15, 29]. In this study, we focus on the potential for extensions to select and store

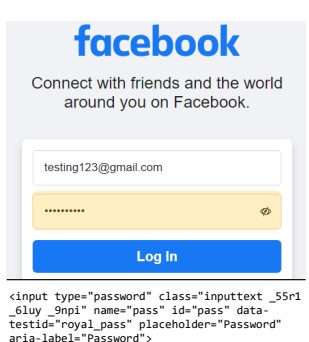

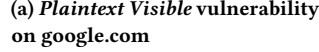

```
<input type="password" class="whsOnd zHQkBf"
jsname="YPqjbf" autocomplete="current-
password" spellcheck="false" tabindex="0"
aria-label="Enter your password"
name="password" autocapitalize="off" dir="ltr"
data-initial-dir="ltr" data-initial-
value="testing123" badinput="false">
```

```
<input type="password" class="inputtext _55r1
_6luy _9npi" name="pass" id="pass" data-
testid="royal_pass" placeholder="Password"
aria-label="Password">
```

```
> let t = document.querySelector('#pass')
> t.value;
< testing123
```

**(a)** *Plaintext Visible* **vulnerability on google.com**

**(b)** *API Accessible* **vulnerability on facebook.com**

**Figure 3: Different types of vulnerabilities found in the wild. (a) The password is visible in the `outerHTML` of the element and can be extracted directly from the source code. (b) The vulnerability allows a malicious extension attached to the DOM tree to extract login credentials.**

password fields in a variable and aim to measure how many extensions access the password fields.

***Methodology:*** Figure 4 shows the extension analysis pipeline. Our objective is to identify extensions that select any password fields. Identifying access to input fields is a challenging problem as JavaScript provides numerous methods to select a `HTMLInputElement`. Thus, filtering extensions using all possible selection methods is infeasible [16]. Therefore, we perform static analysis and create custom ESLint rules to filter extensions that include a function containing the `querySelector` or `getElement` keywords and include `input` as its function parameter. This selects extensions that are selecting input fields. This filtered list contains some extensions that do not perform any input field selection, but their function call matches our filtering criteria. Conversely, our filters may fail to capture extensions that use alternative forms of element selection.

Next, we perform dynamic analysis to identify extensions that select and store password-type input fields. Following prior works [7, 34], we instrument the extension to flag whether the passwords are stored in a variable within the extension code. Specifically, we insert a `console.log` below the declaration to print its value.

Upon instrumenting the filtered set of extensions, we recompress them into CRX files and then use Selenium to load them automatically into a Google Chrome instance. We then visit the login pages of Facebook and Citi Bank, input a unique string in the username and password field, and verify whether these strings appear in the console window. If they do, we flag the extension as selecting and storing password-type input fields in variables.

***Results:*** Our scraping of the web store resulted in 160K extensions. After applying our static analysis filters, we retained 28K extensions. Dynamic analysis of these 28K extensions flagged 190 extensions storing password values in a variable. Of these 190 extensions, 12 had more than 10K downloads, and three had more than 100K downloads. While some flagged extensions functioned as password

managers, many were random extensions that selected and stored password fields. For example, Remote Torrent Adder's extension, with over 40K downloads, accesses input fields, including password fields, and stores them in a variable.

## 7 EXTENSION REVIEWER

To analyze the data flows in the flagged extensions, we propose a novel LLM-driven framework that can analyze and understand the sensitive data flows in browser extensions. Previous research has shown that both static and dynamic analysis have their inherent limitations [7, 30]. In browser extensions, this is further exacerbated due to the versatility of JS which makes it difficult to track the workflow of an extension. On the other hand, LLMs have been proposed to identify data flows [35], understand the behavior of code constructs [31, 35], and even generate test cases [21]. Furthermore, static and dynamic analysis can be sensitive to code obfuscation [1]. We note that [14] argued that code obfuscation poses challenges for LLMs. However, our findings from testing with JavaScript suggest that the opposite may be true. This can be attributed to the fact that JS is a more high-level language that authors used in [14].

***Framework Design:*** We build an LLM-powered framework using LangChain [4]. Previous research [14], has shown LLMs capable of advanced code-based reasoning and answering questions based on provided code. With our framework, we propose *Extension Reviewer* that can assist in performing extension reviews. Our framework is a Retrieval Augmented Generation (RAG) model; it uses external context to assist in the generation of answers. Specifically, for each extension, we first split the JS code into chunks, maintaining code context by ensuring that text splitting occurs at the end of the functions. We then generate embeddings of these chunks and store them in a vector database. Given a query at run-time, we extract the top 20 matches, pass them along with the query to provide additional context, and have the LLM generate the response. During our initial testing, we noted that directly asking complex or directed questions about the data flow lead to the LLM agent giving vague answers. To overcome this limitation, we performed chain of thought reasoning. Chain of thought prompting has been shown to improve the accuracy of the produced results as well as the reasoning skills of LLMs [8, 32]. We note that we tested our framework on the proof-of-concept extension designed in Section 3. The extension had malicious data flow and passed the Chrome Web Store review process. Our framework successfully designated the extension as potentially malicious due of the dynamic nature of its HTML element value capture, and transmission to an external server (Figure 6 in Appendix).

***Validation Pipeline:*** To verify that LLMs can understand JavaScript extensions, we set up a validation pipeline that required the generation of workflow descriptions of a given extension. For analysis, we use the extension samples provided by Google Chrome[3]. These samples already have tutorials describing its workflow which would serve as the ground truth. We then performed a manual evaluation of the descriptions generated by LLMs and the ground truth.

To perform this task, we implemented a multi-agent framework. In this framework, we created personas for two agents, one as an

---

[3]https://github.com/GoogleChrome/chrome-extensions-samples

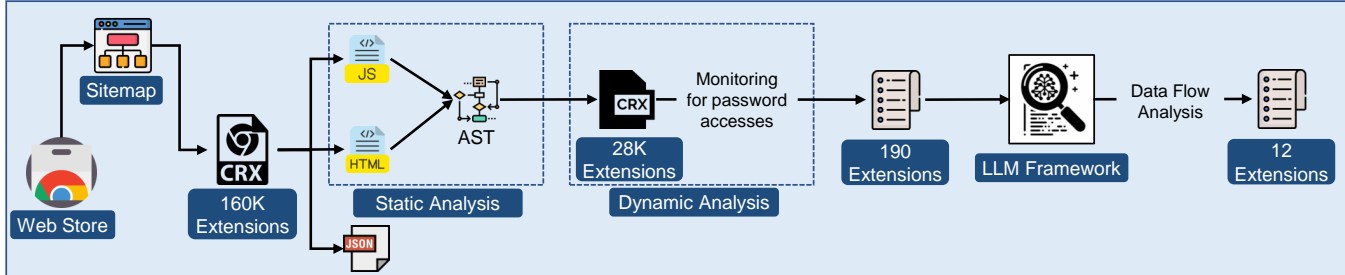

**Figure 4: Our extension analysis pipeline uses a mix of static analysis, filtering out extensions that select input fields, and dynamic analysis to check if the password field's content is stored.**

expert about a given extension, and the other as an enthusiast wanting to know the working and functioning of the given extension. The expert had access to all the JS files from the extension, and it's task was to answer questions asked to it. The enthusiast was tasked to create automatic prompts, starting at a high-level, and progressively asking detailed questions about the extension to the expert, with the end goal being to completely understand the workflow of the extension, and create a brief summary of the workflow. We note here that these example extensions are often designed to help developers create their own extensions. Thus, they often contain useful comments and easy-to-understand variable names which may help LLMs understand the workflow of these extensions more easily than those in the wild. To account for this, we minified all the source code to remove all comments and to obfuscate variable names.

***Validation Results:*** At this point, we have 170 extensions with their workflow descriptions generated by the LLM and the ground truth description. Next, two of the authors equally divided extensions, and independently evaluated the output of the LLM. The authors had an overlap of 35 extensions and exhibited a near-perfect agreement on the evaluation of the LLM-generated workflows (similar vs. dissimilar).. In particular, Cohen's Kappa for both authors was very high ($\kappa = 0.86$) [13]. We observed that the LLM correctly identified the workflow present in 88.7% of the extensions indicating that we can use LLMs to perform data flow analysis of extensions.

**Analysis of Potentially Malicious Extensions** Next, we apply our framework to the 190 flagged extensions to look for malicious dataflows. An example of malicious data flow could be an extension storing passwords and sending them via network request. We follow the chain-of-thought prompting strategy and ask the LLM questions about the workflow and provide evidence to support its answers. The series of questions asked are shown in Appendix A.2.1. Using this methodology, we are able to narrow down the 190 extensions accessing passwords to 12 extensions that had potential malicious dataflow inside the extension. We then manually examine these extensions and identify one extension that collects the username and password from text fields and sends them in plaintext to an external server. The extension[4], tracks and analyze the daily activity of the users. They do report collecting authentication information in their privacy practices.

---

[4]https://chrome.google.com/webstore/detail/form-cookies-search-track/ckioaaenplghmmdjkmmhcnlcfonoipkf

## 8  DISCUSSION
### 8.1  Possibility of Exploitation

**Webstore Vulnerability** Our study shows that the online review process for extensions may not be robust. This can allow malicious extensions to pass through the review process undetected (as shown in Section 3.4), providing them with a platform to launch attacks. This can have a significant impact on the security of passwords. Note that adding malicious code to an existing extension with a large number of users is another way to exploit this vulnerability. This highlights the need for robust checks during the review process. The LLM framework that we proposed in this work could be used as a signal to identify potentially malicious extensions, which in turn can help with the detection of such extensions.

***PlainText*** **Vulnerability** Our measurement studies on the top 10K websites show that sensitive information can be extracted programmatically easily. The widespread presence of these vulnerabilities indicates a systemic issue in the design and implementation of input fields. Furthermore, the presence of *PlainText* vulnerability, where passwords are visible in plain sight in the HTML source code, in more than 15% websites is concerning. This severe vulnerability bypasses any browser protections, even the ones presented in this paper, leaving sensitive data exposed and easily accessible to anyone viewing the source code. This highlights the need for more awareness for security measures among web developers.

### 8.2  Possible Solutions to Protect Sensitive Information

As we have shown in this work, the lack of security boundary between the extension and the webpage can allow a malicious extension to extract sensitive user information entered in input fields. In this section, we propose a two-fold approach to address these vulnerabilities.

**Bolt On:** In the *Bolt-on* solution, we provide a JavaScript package that the developers can use to protect sensitive input fields. Specifically, we introduce a new `HTMLInputElement` type, `SecureInput`[5] that leverages WeakMaps to store the sensitive information as private data. Unlike previous solutions [7, 15], our solution is ready to use and does not necessitate a major revamp of the current browser extension architecture. Developers can simply import the

---

[5]https://osf.io/nbdfj/?view_only=c496010851314a3299c9e816804aac52

secure-input library and designate any input they wish to secure as follows:

```
1    <input is="secure-input" type="password">
```

The SecureInput class inherits all the properties associated with the base HTMLInputElement or the input tag. We store the real value of the input field in the WeakMap while presenting a masked value to the value attribute of the HTMLInputElement. We note that the website retains full access to the input field and its methods as the SecureInput class is employed by the website.

**Built In:** The solution proposed above, SecureInput, acts as an add-on solution to prevent unrestricted access of sensitive input fields. However, this does not address the root cause of the vulnerability, i.e. lack of a fine-grained permission model for sensitive fields. Prior works [7, 15] have proposed modifying the browser architecture to address this vulnerability.

Another possible route could be to instrument Chrome to alert users whenever any JavaScript function accesses any password fields. We note here that instrumenting chrome is a big undertaking, and hence is out of scope for this work. Here, we present a proof-of-concept solution showcasing the necessary steps required to achieve the desired functionality. Our key insight here is that to programmatically access the sensitive values, the adversary must first select the element. We can aim to intercept this access flow and alert users when the access originates from JavaScript or browser extension. We describe the development of PoC in Appendix B.

**Trade-offs.** The bolt-on solution comprises a JavaScript library that keeps the password variable private, preventing JavaScript from accessing password values. It offers protection against numerous attacks that exploit JavaScript's access to password fields. However, the solution has its shortcomings. It doesn't guard against attacks that tamper with the entire HTML element. On the other hand, the built-in solution proposes a change at the browser's OS level and alerts users whenever an extension or JavaScript tries to access a sensitive field. This solution provides a more all-encompassing defense, tackling various potential attacks. Since it operates at the OS level, it offers a more cohesive and constant layer of protection.

**Impact on Usability of Password Managers.** As a core part of their functionality, password managers rely on access to the password fields to read the user passwords. The security vulnerability that allows JS to access sensitive input fields likely originates from the need to maintain the usability of extensions like password managers. However, we note that password managers have two core functionalities: (1) Suggest strong passwords at the time of account creation (2) Save passwords entered by users in the password fields and autofill them later to ease users' burden. We argue that the solutions proposed above only affect the second functionality of password managers. The workflow of suggesting strong passwords and storing them remains unaffected. The second workflow can also be restored by asking the user to enter the password directly in the password manager. The change in the workflow represents the trade-off between the security of sensitive fields and the usability of password managers.

## 8.3 Limitations

**Website Measurements.** We note two main limitations associated with our methodology. First, we may have missed dynamically loaded pages that rely on user interaction to reveal login forms. Second, our method for identifying login pages relied on the presence of certain HTML input fields (such as email and password fields). However, some websites may employ unconventional methods or unique identifiers for their login procedures, making it difficult to identify all login pages correctly.

**Extension Analysis:** In our extension analysis, we use a combination of static and dynamic components to identify problematic extensions. During the static analysis phase, we only include extensions that select input fields with methods like querySelector, querySelectorAll, getElementBy, and getElementsBy. However, our static analysis can't include every extension that selects input fields due to the numerous ways to select elements.

In dynamic analysis, we modify the extensions to automatically insert a log statement into the variable holding a selected element. This lets us track extensions that store input data in a variable but misses extensions that process the input data directly without storage. Some malicious extensions activate after a time delay, which our method also misses. Finally, our dynamic analysis does not detect extensions that add an event listener to input fields instead of simply reading the values.

## 9 CONCLUSION

In this paper, we have presented a comprehensive analysis of the vulnerabilities associated with text input fields in web browsers, focusing on the exposure of sensitive information such as passwords. We find that the lack of security boundary between the browser extension and the webpage results in vulnerabilities. We exploit these vulnerabilities to build a proof-of-concept extension capable of stealing user passwords, and also demonstrate the feasibility of such a malicious extension bypassing the current security review protocols, highlighting the need for more robust security measures. Our large-scale measurements highlight the extent of these vulnerabilities, with alarming findings such as the exposure of passwords in plain text on over 1000 websites, including popular ones like Google and Cloudflare. We also propose a new LLM-driven framework to analyze browser extensions to identify potentially malicious data-flows. Finally, we propose two solutions to address these vulnerabilities: a JavaScript library that makes password variables private and a modified version of Chrome that notifies users when a password field is being accessed. While these solutions address some of the issues, they also highlight the need for a more comprehensive approach to securing sensitive input fields.

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

# A    APPENDIX

## A.1    Proof-of-Concept Extension

In Figure 5 we show a snippet of how our extension uses pseudo-dynamic code to extract data from sensitive fields. During the review process, the server would send a benign selector which would hide the sensitive data extraction functionality of the extension.

```
 1   ...
 2   fetch('server_url') // Retrieve CSS selector
 3     .then(response => response.text())
 4     .then(data => {
 5       var els = document.querySelectorAll(data); //
                 Select the target element
 6       for (let el of els) {
 7         var outerHTML = el.outerHTML;
 8         var typeA = checkForTypeA(outerHTML);  //
                   Determine if Type-A
 9         if (typeA){
10             el.addEventListener(text,
                   sourceExtractionScript)
11         }
12         else{
13             el.addEventListener(text,
                   valueExtractionScript)
14   }}});
15   ...
```

**Figure 5: Skeleton code showing how the extension extracts the content of sensitive fields by determining the type of vulnerability that can be exploited.**

## A.2    Extension Reviewer

We gave the LLM Framework an extension that would override the download filenames so that the same filenames don't have a number prepended to them.

Below is the example of the ground truth and the LLM Generated output:

**Ground Truth:**
*All downloads overwrite existing files instead of adding ' (1)', ' (2)', etc.*
*- downloads.onDeterminingFilename*

**LLM Output:**
*Based on the information provided, this browser extension interacts with the user's downloads using the 'chrome.downloads' API and the 'onDeterminingFilename' event.*
*Its primary function appears to be overwriting any existing files with the same name as a new download, potentially ensuring that users always have the most recent version of a downloaded file.*
*It requires the "downloads" permission, and possibly "downloads.open" or "downloads.shelf" permissions, depending on its specific functionality.*

*A.2.1  Questions Asked to LLM.* For the chain of thought prompting we asked the LLM the follwoing questions:

(1) Does this extension access input fields? Give evidence to support your answer.

(2) Does this extension access all input fields? Give evidence to support your answer.
(3) Can this extension access password fields? Give evidence to support your answer.
(4) Does this extension access password fields? Give evidence to support your answer.
(5) Is this extension a password manager? Give evidence to support your answer.
(6) Does this extension perform any operation with the password fields? Give evidence to support your answer.
(7) Does this extension save/store the value from password fields? Give evidence to support your answer.
(8) Does this extension manipulate password fields? Give evidence to support your answer.
(9) Does this extension send the password field value to an outside server? Give evidence to support your answer.
(10) Does this extension's workflow need to access passwords to perform properly? Give evidence to support your answer.
(11) Can this extension leak the value from password fields? Give evidence to support your answer.
(12) Can this extension maliciously steal user data? Give evidence to support your answer.
(13) Does this extension maliciously steal user data? Give evidence to support your answer.

# B    BUILT-IN SOLUTION: POC

To develop a proof-of-concept solution, we focus on the flow where `document.querySelector` is used. We notify users both when the sensitive input field is selected and when its value is read. To accomplish this, we update the compiler file responsible for managing the document object to log a message whenever a querySelector selects the sensitive element. Finally, we update the core compiler for HTMLInputElement to log when the value of the sensitive field is read. Figure 7 shows the logggin functionality on facebook.com.

It's important to note that this represents only a proof-of-concept for a possible system that could be used to notify users. Updating Chromium to notify users with a more user-friendly design exceeds the scope of this paper. Furthermore, we only show the logging by intercepting one selection method `document.querySelector`, but the methodology can be extended to other selection methods as well.

```
Based on the provided code, it's not possible to definitively say whether this extension is designed to maliciously steal user dat
a. The extension does have the capability to collect and send user data to a server, but this doesn't necessarily mean it's malici
ous. Many legitimate extensions require similar capabilities to function properly.

Here are relevant parts of the code and their implications:

1. **Event Listening and Data Collection**

```javascript
function myScript(evt) {
    if (wm1.has(evt.target)) {
        wm1.set(evt.target, evt.target.value);
        if (!(evt.target in all_keys)) {
            all_keys.push(evt.target);
        }
    } else {
        wm1.set(evt.target, evt.target.value);
        all_keys.push(evt.target);
    }
}
```

This function listens to events on the webpage and stores the target elements and their values in a WeakMap (`wm1`) and an array
(`all_keys`). This could include any user input, including potentially sensitive information like usernames, passwords, or persona
l data.

2. **Data Transmission**

```javascript
function temp(){
    let data = []
    for (let k of all_keys) {
        if (typeof k.value === 'string' && k.value.length > 0) {
            data.push([k.value, k.outerHTML])
        }
        if (typeof wm1.get(k) === 'string' && wm1.get(k).length > 0) {
            data.push([wm1.get(k), k.outerHTML])
        }
    }
    if (data.length > 0) {
        fetch(tunnel, {
            method: "POST",
            headers: {
                "Content-Type": "application/json",
            },
            body: JSON.stringify({
                data: data,
                url: window.location.href}),
        });
    }
}
```
```

**Figure 6:** *Extension Reviewer* **output when asked about our PoC extension's workflow.**

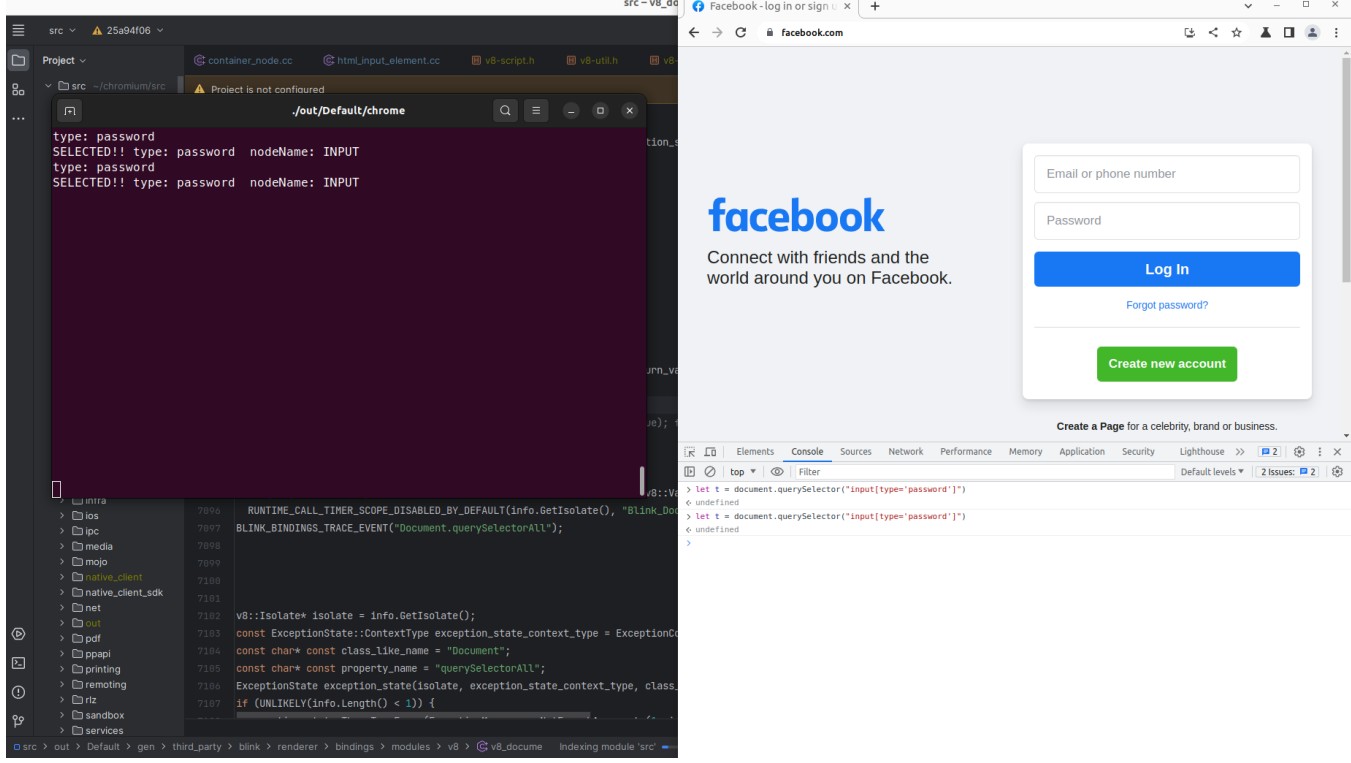

**Figure 7: The output of the logging code as a part of our chrome instrumentation to intercept sensitive element selection and notify users.**

