# OpenReview forum: "Experimental Security Analyses of Access of Browser Extensions Accessing Sensitive Input Fields"
_ACM.org/TheWebConf/2024/Conference — TheWebConf24 Oral_

### Official Review · Reviewer_jVHu · 2023-11-05

**Novelty:** 4
**Technical Quality:** 4

**Review:**

This paper did a fairly comprehensive study regarding the access to the password field by a malicious extension. It first builds a new PoC extension that can access the password field and bypass Google's Manifest V3 policy, and the authors show it was successfully approved by Google's review procedure. Then, it crawled 10K websites and identify 1100 websites exhibit Plaintext Visible vulnerability. Finally, it crawled 160K extensions from Google web store and applied program analysis + LLM to detect extensions that access and store password fields.

The authors' efforts are applaudable as many tasks are carried out under this theme. I have a few concerns.

 1. I feel the authors try to scramble too many things into the 8 pages and leave the technical details not well explained. like the static and dynamic analysis of extensions.

 2. That the authors submit a PoC extension to Google web store can lead to ethical issues. Though the authors try to avoid harming the privacy of the human testers, their time is wasted inevitably.

 3. The PoC extension didn't leak the password to a remote server, but I feel that could be the reason that Google approves it, as no real damage/threat is there.

 4. Though the authors disclose the issues to Google, Google's responses are all negative, without further actions to fix the problem.

 5. I don't see why Plaintext Visible is a problem. If the attacker owns a webpage in different origin, she/he can't access the plaintext field. If the attacker runs an extension, she/he can access the password with other means, e.g., hooking the event handler.

 6. It's nice that the authors consider LLM to further analyze the extensions to determine their intentions, but I don't know how it outperforms the other standard approaches. Static and dynamic analysis seems to be extensible to this task and used as a base to compare with LLM based approach.

7. For "Bolt On" defense, the password is stored in WeakMaps. How is this more secure against the powerful extensions?

8. The authors acknowledge the limitations of methodologies, like not being able to handle the delayed activation of malicious extensions and event listeners. These issues have been discussed by prior works before and they have been used by real-world attackers. I don't think they are small issues that could be ignored.

**Questions:**

1. Is there a more fine-grained result characterization of the measurement study about websites and extensions?

2. Can more details about static and dynamic analysis be given?

**Ethics Review Description:**

A PoC malicious extension is uploaded by the authors to Google web store to demonstrate the attack effectiveness. The human tester's time is wasted during manual review.

**Ethics Review Flag:**

Yes

**Reviewer Confidence:**

3: The reviewer is confident but not certain that the evaluation is correct

**Scope:**

4: The work is relevant to the Web and to the track, and is of broad interest to the community

---

### Official Review · Reviewer_FtSR · 2023-11-23

**Novelty:** 4
**Technical Quality:** 4

**Review:**

The paper presents several experiments relating to Chrome browser extensions, in terms of their access to password fields
- Demonstrate that Chrome Web Store does not detect a password-stealing extension
- Demonstrate that many popular websites process passwords in a way that make them available to extensions
- Test extensions for malicious behaviour through a combination of static & dynamic analysis and LLM-driven checks

I think this is an important topic, since there are clearly malicious attackers exploiting the vulnerabilities opened up by browser extensions. I however am not sure of the threat model that this work is assuming. Extensions are very powerful, and it seems unavoidable that a malicious extension can cause significant harm. This is backed up by Google's response to the authors, essentially arguing that this is expected behaviour. On the other hand, Chrome does impose security constraints on extensions, particularly Manifest V3.

I don't see any significant ethical issues, but under some reasonable definitions of human subject research this did experiment on member(s) of the Chrome Web Store manual testing team. It would be good to see an explicit statement of whether IRB (or equivalent) approval was sought and if not, why not.

I understand why during the test the data-receiving server was deactivated, but could this have disrupted the checks that Google perform? If they are looking for sensitive data being sent, the failed connection might cause the checks to conclude the extension is safe, but had the server been running it would have given the checks more opportunity to detect the sensitive data egress.

The assumption seems to be that passwords are sensitive, which is probably true but what about other sensitive data (health data, banking information etc.) Can these be captured too?

The LLM evaluation seemed a bit out of place with the rest of the paper, and there are some methodological issues. For example the paper describes how comments were removed from the extensions, but could the LLM have been trained on the commented code (available on GitHub)? Also there was suspicious behaviour found, but how much malicious behaviour was missed. Both are important considerations for assessing accuracy.

**Questions:**

- Is there an agreed threat model for which browser extensions should be evaluated against? How does the work presented in the paper align with this threat model.
- Was an IRB or similar consulted. If not, please explain why this was not considered necessary.
- Was the LLM test methodologically sound?

**Reviewer Confidence:**

3: The reviewer is confident but not certain that the evaluation is correct

**Scope:**

4: The work is relevant to the Web and to the track, and is of broad interest to the community

---

### Official Review · Reviewer_5G38 · 2023-11-23

**Novelty:** 5
**Technical Quality:** 5

**Review:**

-  The paper motivates the problem of security issues in browser extensions (accessing sensitive fields, and some websites saving the passwords in plain text) by building PoC hybrid extension attack using techniques from static and dynamic code injection. To address this issue, the authors proposed an LLM-based extension reviewer and discussed ways to address the problem in more detail.

Strengths:
S1: Extensive analysis of browser extensions on the WebStore to find their vulnerabilities that might potentially leak sensitive information such as passwords or SSNs.
S2: Novel hybrid malicious browser extension that bypasses the Webstore review highlighting the necessity for more rigorous review procedure and strategy for browser extensions.
S3:  LLM-based extension reviewer and thoughtful potential approaches (bolt-on: JS library containing secure-input element) and future directions (built-in option: change at the browser's OS level to alert users about malicious acts of extensions) to address the vulnerabilities.

Weaknesses:
W1: Lack of details in the performance analysis of LLM-based browser extension reviewer.
W2: Lack of comparisons with other existing baselines.

**Questions:**

1. It would be great if the authors could provide more details about the performance of LLM-based extension review. For instance, with the extension reviewer, the authors were able to narrow down the 190 extensions to 12 that had potential malicious dataflow. What about the remaining extensions? Were there cases where there was a potential malicious dataflow, but the LLM failed to detect it?

2. How would existing work in extension analysis perform compared to the proposed LLM extension reviewer? It would be helpful to provide comparison of the proposed extension reviewer against the baselines.

3. When it comes to the proposal of LLM-based solution and other potential solutions (bolt-on, and built-in), I feel like a stronger link would be helpful. In which cases should the LLM-based extension reviewer be preferred? Where should it be built and what overhead (from the user/developer perspective) could it cause compared to bolt-on and built-in solutions?

4. It would be interesting to see how the findings with the Google Chrome-based extensions can be generalized or extended to other famous browsers such as Apple's Safari, Microsoft's Edge, and more privacy-preserving browsers such as Brave.

5. Section 7: Validation Pipeline. How was the minification of source code done? What percentage of variables and comments are obfuscated/removed?

6. Section 8.2: Impact on usability of password managers. It seems like the proposed approaches still affect the important usability aspect of password managers, where the user will have to type in the passwords to the password manager every time. Given this, can the authors (at the beginning of the Intro) argue that the provided potential solutions can maintain the usability of password managers?

Minor comments:
- 2.3: typo: tranformer→ transformer-based models
- 3.1: typo in the title: Priviledges →  Privileges

**Reviewer Confidence:**

3: The reviewer is confident but not certain that the evaluation is correct

**Scope:**

4: The work is relevant to the Web and to the track, and is of broad interest to the community

---

### Official Review · Reviewer_5XnK · 2023-11-24

**Novelty:** 5
**Technical Quality:** 5

**Review:**

This paper presents an empirical study on the security risks posed by browser extensions. The authors develop a proof-of-concept extension disguised as a ChatGPT plugin that can extract users’ passwords. They find that the extension passes the Chrome WebStore review process, indicating weaknesses in the current review mechanisms. They analyze the login pages of the top10K domains and identify that password values can be accessed using their extension. They also perform static and dynamic analysis of19K extensions on the WebStore and identify190 potentially malicious extensions that access password fields. To analyze the behavioral data flows of these extensions, they propose a novel LLM-driven framework called Extension Reviewer. The paper discusses the prevalence of malicious extensions in the WebStore and the need for more robust verification systems. It also highlights the PlainText vulnerability where passwords are visible in HTML source code. The authors suggest countermeasures such as a JavaScript package for website developers to protect sensitive input fields and a browser-level solution to alert users when an extension accesses sensitive input fields. Overall, the paper emphasizes the importance of improved security measures to safeguard sensitive user information online.

Pros:
1. The paper is well-written and easy to understand.
2. The paper points out a privacy vulnerability of current browser extensions.
3. A novel LLM-driven framework named Extension Reviewer is developed for detection.

Cons:
1. Extension Reviewer is not compared to existing methods [1].
    *  The authors state the extension reviewer is proposed to solve the inherent limitations of both static and dynamic analysis. However, no detailed experiments are conduct to validate the point.
2. Some existing works also point out there is privacy leakage in browser extension [1]. What is the relationship between the risks point out in the previous work and the risk in this work? Can previous analysis method identify the risk found in this paper?
3. I suggest the authors to report the risk to the service provider of browsers to avoid being adversarially exploited.

[1] Mystique: Uncovering Information Leakage from Browser Extensions

**Questions:**

None

**Reviewer Confidence:**

3: The reviewer is confident but not certain that the evaluation is correct

**Scope:**

4: The work is relevant to the Web and to the track, and is of broad interest to the community

---

### Official Review · Reviewer_9EYZ · 2023-11-24

**Novelty:** 2
**Technical Quality:** 4

**Review:**

The paper analyzes the security implications associated with browser extensions, shedding light on potential risks of unauthorized access to sensitive user data. Through the creation of a proof-of-concept extension, the authors expose vulnerabilities in the Chrome WebStore review process, highlighting gaps in current security protocols. The empirical study examines the login pages of top websites and over 160,000 Chrome Web store extensions, offering substantial insights into the prevalence of vulnerabilities. Notably, the paper identifies that 28,000 extensions possess permissions to access sensitive fields, while 190 extensions store password fields in variables.


The paper's contributions in comparison to existing works raise concerns. While the development of a proof-of-concept extension and the introduction of the Extension Reviewer framework highlight potential vulnerabilities, the paper lacks a sufficiently clear distinction of its novel contributions compared to prior research in the field.
The paper could benefit from a more explicit articulation of its novelty and unique contributions in comparison to existing research in the field. Moreover, it is essential for the authors to finalize the related works section and incorporate closely related papers. For example, a citation to "Dolière Francis Somé. EmPoWeb: Empowering Web Applications with Browser Extensions. S&P 2019" is missing.

**Questions:**

- Can you discuss the conceptual distinctions between your method and those employed in previous research?

-  Have you conducted an evaluation comparing your method with previously proposed approaches?

**Ethics Review Description:**

The paper raises ethical concerns as the authors designed an extension with the capability to potentially compromise sensitive user information. However, this concern was addressed by the authors, who respected the  ethical considerations in the design of their study as described in Section 3.4.

**Reviewer Confidence:**

2: The reviewer is willing to defend the evaluation, but it is likely that the reviewer did not understand parts of the paper

**Scope:**

4: The work is relevant to the Web and to the track, and is of broad interest to the community

---

### Decision · Program_Chairs · 2024-01-22

**Decision:**

Accept (Oral)

**Comment:**

We assess that it is important for the topic of this paper to be included in the conversations of TheWebConf 2024. As a result, and after careful review of the reviewers assessment and the area chair's recommendation (below), we recommend this paper to be accepted. We ask the authors to improve the paper in response to feedback for camera-ready. We particularly underline the importance of further addressing the ethical questions raised and mentioning the IRB response in the camera-ready version.

"This paper presents an empirical study on the security risks posed by browser extensions. They do a ChatGPT plugin that can extract users' passwords and that passes the Chrome WebStore extension review process. Then they analyze the login pages of top 10K domains and identify that password values can be accessed using their extension. They also perform static and dynamic analysis of 19K extensions on the WebStore and identify that 1% of them have the potential to (maliciously) access password fields. Overall, this work stresses the importance of improved security measures to safeguard sensitive user information online.

 Strengths:

 - This research address an important problem and exposes privacy vulnerabilities of Chrome extensions.
 - Proposes a novel hybrid malicious browser extension that bypasses the current Chrome Webstore review.
 - Gives a novel LLM-driven framework is developed for this task.
 - The paper is well-written and easy to understand.

 Weaknesses:

 - There is no comparison to other solutions, if any.
 - The authors do not validate that the extension reviewer is proposed to solve the inherent limitations of both static and dynamic analysis.
 - Lack of details in the performance analysis of LLM-based browser extension reviewer.
 - Some relevant previous work is not cited

 Scope: 4; Novelty: 4; Quality: 5"